# Clay-Catalyzed Ozonation of Organic Pollutants in Water and Toxicity on *Lemna minor*: Effects of Molecular Structure and Interactions

**DOI:** 10.3390/molecules28010222

**Published:** 2022-12-27

**Authors:** Eric Noel Foka Wembe, Amina Benghafour, David Dewez, Abdelkrim Azzouz

**Affiliations:** 1Nanoqam, Department of Chemistry, University of Quebec at Montreal, Montreal, QC H3C 3P8, Canada; 2École de Technologie Supérieure, Montréal, QC H3C 1K3, Canada

**Keywords:** clays, montmorillonite, atrazine, bisphenol A, diazinon, diclofenac sodium, adsorption, catalytic ozonation, ecotoxicity, oxidative degradation, *Lemna minor*

## Abstract

The use of clays as adsorbents and catalysts in the ozonation of organic pollutants (Atrazine, bis-Phenol A, Diazinon, and Diclofenac sodium) allowed simulating their natural oxidative degradation in clay soils and to evaluate the ecotoxicity of mixtures partially oxidized on the species *Lemna minor*, a biodiversity representative of plants in the aquatic environment. Kinetic data showed that the adsorption of organic pollutants on clay particles obeys the pseudo-second-order model, while the adsorption isotherms satisfactorily fit the Langmuir model. Adsorption reduces the dispersion of the organic pollutant in the environment and prolongs its persistence and its natural degradation probability. Measurements of the Zeta potential and particle size as a function of pH demonstrate that the catalytic activity of clay depends on its cation, its silica/alumina ratio, and therefore on its permanent and temporary ion exchange capacities. These factors seem to govern its delamination and dispersion in aqueous media, its hydrophilic-hydrophobic character, and its porosity. Tests conducted on *Lemna minor* in contact with ozonation mixtures revealed that the toxicity could be due to pH decrease and to the toxicity of the intermediates yielded. Ecotoxicity would depend on the structure of the organic molecules, the chemical composition of the clay surface and ozonation time, which determines the oxidation progress. These results are of great importance for further research because they allow concluding that the negative impact of the persistence of an organic molecule in clay-containing media depends on the type and composition of the very clay mineral.

## 1. Introduction

Human activities are sources of soil and water pollution. These activities release various types of pollutants into nature such as greenhouse gases and SMOG, pesticides and insecticides, household and industrial-agricultural wastewaters. The latter may contain pharmaceutical products, plastics, endocrine disruptors, antibiotics and biological pollutants and others [1,2,3,4]. Pesticides and drugs such as Diazinon and Diclofenac along with their derivatives are particularly hazardous for both human health and biodiversity [5,6,7,8,9,10,11,12]. For instance, Diazinon is considered as a “moderately hazardous” class II pesticide displaying a lethal dose for humans of approximately 90–444 mg kg^−1^ with a maximum acceptable dose of 1 ppm and a toxicity dose to aquatic organisms of 350 ng L^−1^ [13,14,15].

A pollutant dispersed in soils and waters may undergo various physical-chemical transformations among which the oxidative degradation processes are the most predominant on air-water-soil interfaces [1,4,16,17,18,19]. If soluble in soil waters, organic pollutants can diffuse and contaminate surface and ground waters [14]. Once interacting with air and/or oxidizing agents in soils and clay-containing media, their unavoidable degradation gives rise to a variety of more or less harmful derivatives that can spread in the environment by volatilization and/or solubility in water.

To date, many techniques involving chemical, physical, biological or thermal treatments have been tested more or less successfully for organic molecule removal [20,21,22,23]. All these techniques turned out to be unviable because they involve high investment costs and energy consumption, especially from fossil sources which produce polluting gases. In addition, these methods do not take advantage of the intrinsic capacity of soils to self-decontaminate and self-regenerate. Among the so-called Advanced Oxidative Processes [24,25,26,27,28,29,30,31], clay-catalyzed ozonation could be a promising green strategy for this purpose [2].

The simultaneous action of clay that acts as a soil component and ozone is whose action is similar to that of natural oxidizing agents allows simulating natural oxidative processes. Their more or less slow evolution in time determines the persistence of organic pollutants in nature. The negative impacts on biodiversity and human health varies according to their molecular structure and interactions with key-components in soils and waters. Thus, the very target of this research resides in demonstrating that incompletely oxidized organic pollutants induce various ecotoxicity and that the latter can be predicted by the type of organic pollutant and by the host clay-rich media. In other words, the mere presence of clay minerals can differently reduce ecotoxicity and could be beneficial for natural remediation without sophisticated and costly technologies.

Adsorption is undoubtedly the main phenomenon occurring between pollutants and soil particles [32,33,34]. Some soil components such as dense and microporous silicas are fairly hydrophobic and promote interactions that favor the retention of organic species on solid particles, thereby prolonging their residence time and raising their degradation probability. Clay minerals are expected to also involve hydrophilic, Lewis acid-base and electrostatic interaction that need to be investigated. In this regard, a major challenge first resides in correlating the behavior of three organic molecules, namely bis-Phenol A (BPA), Diazinon (DAZ) and Diclofenac sodium (DCF) in the presence of various clay minerals and their toxicity on *Lemna minor*. This plant biodiversity representative is known to be affected by organic pollutants and be used to assess the toxicity of polluted waters [35,36,37,38,39,40,41,42,43,44]. The effect of the evolution in time of the ozonation of four organic species on the toxicity of the reaction mixture on *Lemna minor* will be assessed in the presence of natural clay materials.

The use of clay-catalyzed ozonation was intended to simulate the natural behavior of clay-containing matrices (Soils and turbid aquatic media) for a better understanding of their self-regeneration. For this purpose, crude and acid-activated bentonites and various montmorillonites exchanged with harmless metal cations (Na^+^, K^+^, Ca^2+^ and Fe^2+^) that already occur in most soils were used both as adsorbents and catalysts. Their reuse is as convenient as their very preparation, but their reusability is needed only if justified given their low cost and wide availability.

Such an unprecedented approach allows “mimicking” an accelerated natural oxidative degradation of organic pollutants in soils by using ozone, which is expected to generate similar intermediates as in soils exposed to the sun and air. The results of this work are expected to provide valuable data that allow predicting the degradability of organic pollutants, their intrinsic ecotoxicity, and those of their derivatives according to the soil structure and composition.

## 2. Results and Discussion

### 2.1. Clay Behavior in Aqueous Media

As expected, the aqueous solutions of the different organic species investigated showed different intrinsic pH (pH_S_) (Appendix A). As a general feature, all organic molecules induced a pH decrease starting from the initial value of pure distilled water (pH = 6.59), slightly more pronounced down to 4.78 for DCF. This is due to their specific pKa values and acid-base interactions with water molecules. Addition of different amounts of clay minerals and pH adjustment at well-established initial values (pHi) revealed a pH decrease after 10 min impregnation, except for BPA and DAZ at high initial pH (pHi = 12). The most pronounced pH decreases were registered with the most acidic clay catalysts (Fe(II)Met and HMt-1) in ATR and DCF solution. This pH decrease must be due to the rise of specific Clay:Water and Clay:Substrate interactions. The pseudo-plateau illustrated by slight pH increase upon increasing adjusted pH from 3 to 9 (Figure 1) suggests the occurrence a buffering effect.

The fact that no buffering effect was noticed in the absence of clay or organic species must involve clay interactions with the organic substrate and/or water. Such interactions may be induced by the “out-of-plane” (pKa 5.6) and “in-plane” (pKa 8.5) silanols within this pH range [45] and are expected to govern the surface charge that determine the clay dispersion and particle size in the aqueous media. As expected, the general tendency is a decrease in clay particle size with increasing Zeta potential (ZP) from −20 mV to −100 mV, due to increased surface charge that promotes interlamellar repulsion forces, suggesting the predominance of clay-water interaction (Figure 2a).

Nonetheless, adsorption appears to reduce the Zeta potential by generating layers of organic compound. The latter are not only expected to shade the repulsive surface charge but also promote organophilic interaction that leads to clay lamella aggregation and loss in porosity and accessible surface. This is well supported by a decreasing tendency of the Zeta potential with increasing adsorption (Figure 2b). This shading effect of adsorption should progressively fade upon enhanced ozonation that cleans the adsorptive-catalytic surface up to certain equilibrium. Such an equilibrium is supposed to be governed by the affinity of the organic substrate towards the clay surface and its reactivity towards adsorbed ozone. This barely detectable decreasing tendency illustrated by the high dispersion of the experimental data must be due to various adsorption-ozonation equilibria for each clay-substrate pair. Deeper insights in adsorption kinetics could explain, at least partly, the surface interactions involved.

### 2.2. Adsorption Kinetics on Clay Surface

Negligible to even no adsorption was noticed at pH 12 in spite of the highest ZP values obtained at this pH level (Table 1). This can be explained in terms of repulsion forces between the negatively charged clay surface and highly deprotonated organic substrate.

Adsorption equilibrium was attained after impregnation times ranging between 7 and 10 min. Specific retention rate were registered depending to the clay-substrate couple reaching maximum values of ca. 80% for the couple ATR-Fe(II)Mt, 60% for ATR-HMt-1 and DCF-Fe(II)Mt) and 55–56% for DAZ-HMt-1 (Appendix A). BPA showed the lowest adsorption rates on all clay minerals, while KMt and CaMt displayed low affinity towards the organic substrates even almost negligible for ATR.

Attempts to investigate the adsorption kinetics were achieved applying the pseudo first and second order models by respectively plotting as functions of time (Equations (1) and (2)).
(1)$$Log\left(q_{e}-q_{t}\right)=Log\left(q_{e}\right)-\frac{k_{1}}{2303}t$$
(2)$$\frac{t}{q_{t}}=\frac{1}{k_{2}q_{e}^{2}}+\frac{1}{q_{e}}t$$
where (***q_e_***) and (***q_t_***) account for the equilibrium and instant of adsorbed organic substrate. A first overview revealed that linearity was obtained with both models but for different Clay-Adsorbate couple (Appendix A).

At first glance, the highest and closest correlation coefficients (R^2^) to unity were obtained for the pseudo-second order kinetics, except for DAZ adsorption on Fe(II)Mt (R_1_^2^ = 0.9332) (Table 2). These data indicate that diazinon adsorption on Fe(II)Mt predominantly occurs via purely physical interaction unlike the other adsorbate-adsorbent couples. The close values of the correlation coefficients for DCF (0.9303 and 0.9463) suggests that adsorption on Fe(II)Mt obeys a mixture of the two kinetics models involving both physical and chemical interactions. A qualitative comparison revealed higher pseudo-1st order constant (0.0852 versus 0.0790) suggesting a faster physical adsorption of DCF on Fe(II)Mt as compared to that of DAZ.

### 2.3. Intra-Particle Diffusion Model

An overview of the dependence ***q_t_ = f*(*t*^1/2^)** [46] revealed the occurrence of successive domains of linearity with progressively decreasing slopes except for BPA whose adsorption kinetics does not seem to clearly fit the model (Figure 3). ATR and DAZ adsorption was characterized by two intra-particle diffusion steps on both clay adsorbents. DCF adsorption on Fe(II)Mt showed two diffusion steps after 60 min and three steps in the presence of HMt-1.

Higher intra-particle diffusion constants were obtained in the first linearity range ((K_1_) for ATR-Fe(II)Mt (10.108), DCF-HMt-1 (6.7404) and DAZ-HMt-1 (5.8454) as compared to the second (K_2_) linearity for all organic substrates (Table 3). Such a phenomenon was somehow expected, and it would correspond to a diffusion of the organic molecule between bare grains more or less aggregated of clay adsorbents followed by diffusion inside organo-clay clusters (Figure 1).

The adsorbate cover is supposed to weld the grains together mainly by hydrophobic interactions (London Force) and/or hydrophilic hydrogen bridges depending on the molecular structure. Such interactions generate an additional and rigid porosity that attenuate the further diffusion of unadsorbed molecules towards internal sites. This suggests the occurrence of “strong” adsorption sites on the solid surface that causes rapid adsorption during the first minutes (first stage) followed by adsorption on progressively weaker sites.

### 2.4. Adsorption Equilibrium

The adsorption equilibrium was studied by applying Langmuir’s (Appendix A) and Freundlich’s (Appendix A) models, respectively, based on purely physical adsorption and adsorption accompanied by a chemical reaction. Here also, clear and high linearity were registered for ATR, DAZ, and DCF adsorption on both Fe(II)Mt and HMt-1 but not for BPA.

The highest values of the correlation coefficient were obtained when applying Langmuir’s model to the adsorption of DAZ on both adsorbents (0.9992, 0.9962, respectively) and a lesser extent that of ATR and DCF on HMt-1 (0.9748 and 0.9563, respectively) (Table 4). This is a precise indicator of the occurrence of physical adsorption. As expected, BPA adsorption on both clay materials showed the lowest R^2^ values for both models. A possible explanation should arise from the specific behavior of the hydroxyls of both phenolic groups that are assumed to induce competitive and pH dependent H-bridge interactions with water molecules and clay surface.

It is worth noting that appreciable values of the correlation coefficient were also obtained when applying Freundlich’s model to the couples ATR-HMt-1 (0.9375), DAZ-Fe(II)Mt (0.9756), DAZ-HMt-1 (0.9594), DCF-Fe(II)Mt (0.9912), DAZ-HMt-1 (0.9178). These values clearly demonstrate the marked contribution of a chemical adsorption of ATR, DAZ, and DCF. This should involve at least the retention of the induced ammonium groups on the nitrogen atoms in slightly acidic media via cation exchange.

### 2.5. Non-Catalytic Ozonation

The first 5 min of non-catalytic ozonation of all organic compounds gave rise to new adsorption bands between 205 nm and 230 nm in the UV-Vis spectra of the reaction mixtures (Figure 4). Except for DCF, these new bands increased in intensity during ozonation due to the progressive enhancement of the formation of oxidized derivatives. The latter are known to also absorb in this UV-Vis region, thereby shading the progressive depletion of the parent organic molecules [2,47,48,49,50,51,52,53,54,55].

This is a major shortcoming of UV-Vis spectrophotometry that restricts its use to only qualitative assessment of the ozonation progress. Quantitative determination of the degradation rate can be achieved by Liquid phase chromatography coupled to UV detection (HPLC-UV), as supported by the visible decay in time of the relative peak area (A/A_o_) of all the three organic substrates investigated herein (Figure 5). DAZ non-catalytic ozonation was found to produce the fastest depletion as compared to DCF and BPA, given the marked A/A_o_ decay down to ca. 50% after less than 1 min ozonation and even a total disappearance after 30 min. DCF ozonation resulted in slower A/A_o_ decrease down to ca. 50% after less than 3–4 min ozonation and 97–98% after 30 min, slightly faster than that of BPA (50% after 4–5 min and 90% after 30 min). However, ozonation resulted in a common intermediate denoted as Int-1 but different derivative distribution according to the parent molecules (Figure 6).

Here, the chemical stability and refractory character towards ozone of an organic species cannot be explained only in terms of its initial oxidized state expressed in terms of oxidizing elements to carbon atom ratio ((O + Cl)/C) [41,56,57,58]. Indeed, the relatively higher (O + Cl)/C values of DCF (0.286) and DAZ (0.250) as compared to BPA (0.133) contrast with their reverse reactivity sequence (Table 1). This indicates that the very molecular structure (conjugated double bonds, oxygen atom bonds, nature of the chemical function bearing the oxygen atom, etc.) and other factors must also be taken into account.

### 2.6. Effect of Clay Catalyst Addition

Clay catalyst addition induced a noticeable acceleration of the ozonation process, more particularly for DCF and BPA. Their almost total disappearance of the substrate HPLC-UV peak (less than 1% residual amount) was observed after only 10 min of ozonation in the presence of Fe(II)Mt and 20–25 min with HMt-1 (Figure 4). The paradoxically lower DAZ conversion rates as compared to the non-catalytic ozonation is not due to an inhibiting effect of the catalyst, but rather to a competitive ozone consumption in side-reactions as suggested by the rise of a fourth intermediate (Table 5).

Comparison of the data resulting from adsorption and catalytic tests showed that BPA is mainly removed by Fe(II)Mt-catalyzed ozonation (99.22% including 9.5% by adsorption) and completely degraded by HMt-1-catalyzed ozonation (99.97%). Moreover, more than half of DAZ seems to be eliminated by adsorption on the two clay materials (58.65% and 58.56%, respectively). Significant contributions of adsorption to the total removal rate were also observed for DCF in the presence of Fe(II)Mt and HMt-1 (60 and 43% versus 98.56 and 100%, respectively). Atrazine removal was also found to involve mainly adsorption in the presence of Fe(II)Mt (80.41%) regardless to the catalytic test data which are still under investigation. This strong affinity of ATR towards Fe(II)Mt surface is consistent with the relatively high values of K_1_ (0.1308 mg/g/min1/2) and K_2_ (0.06413 mg/g/min1/2) in both kinetical model as well as with K_F_ (3.3146 mg/g/min1/2) in the Freundlich’s model (Table 4). Here, atrazine adsorption must involve previous N atom protonation followed by cation-exchange, in agreement with our previous statements.

### 2.7. pH Evolution during Ozonation

The gradual pH decrease observed during ozonation of the investigated organic substrates is due to the progressive production of oxidized derivatives that include carboxylic acids (Figure 7). The fastest and most pronounced pH decrease was noticed for HMT-1-catalyzed DCF ozonation. This accounts for enhanced ozonation as supported by the total removal rate of this substrate (Table 1).

It is worth noticing the shoulder in the shape of the pH decrease curves for all organic substrate between 5 and 10 min of non-catalytic ozonation. This can involve two consecutive steps in the formation of acidic derivatives either, i.e., from the parent organic molecules and then from oxidized intermediates or from direct ozonation by gaseous ozone bubbles and then by dissolved ozone. Even if this shoulder was not observed in clay-catalyzed ozonation, adsorption should not be involved, at least for BPA, given its low contributions on both clay materials (9.5 and 0.0%) (Table 5).

### 2.8. Toxicity of Ozonized Reaction Mixtures

Preliminary assessment of the ecotoxicity towards *Lemna minor* through pigment extraction revealed a decrease in the Chlorophylls *a* to *b* (Chl_A_/Chl_B_) ratio [59,60,61,62] with increasing ozonation time. It is well known today that a decrease of this parameter clearly indicates an alteration of the structural and functional properties of the photosynthetic apparatus. This ratio drooped down to zero for DCF-Fe(II)Mt, DCF-HMt-1 and BPA-HMt-1 mixtures at the instant pH after 10 min ozonation, presumably due to more or less oxidized organic derivatives that exhibit toxicity through their acidity and/or their very molecular structure. Here, the pH decrease in the ozonation mixtures appears to induce specific changes in ecotoxicity depending on the clay catalyst. Indeed, slower effect was obtained after 20 min ozonation of DCF-HMt-1 mixture at pH 6.5 and BPA-Fe(II)Mt at the instant pH after ozonation (Appendix A). This was accompanied by an alteration in the shape, color (presence of necroses) and turgidity.

Maintaining the pH at a constant level of pH 6.5 after ozonation was found to attenuate the acid stress that causes the alteration of the pigment composition, as supported by quasi-constant Chl_A_/Chl_B_ ratios in all aqueous dispersions of clay-substrate mixtures. However, other factors such as changes in iron availability as a micronutrient for plant growth should also be considered. Indeed, acidic pHs are known to produce an excess of cationic iron with potential toxicity, while pHs exceeding 6.5–7.2 induce iron precipitation and detrimental deficiency for plants.

Another ecotoxicity parameter resides in measuring the protein content. Immersion of *Lemna minor* individuals in ozonized DCF aliquots at pH adjusted to 6.5 induced an overall increase in protein content except for DCF-HMt-1 mixture (Figure 8). The latter showed a consecutive and fast depletion of the protein content after less than 10 min. Increase in protein content accounts for protein synthesis despite the oxidative stress produced by the first 5 min of ozonation in the presence of Fe(II)Mt and HMt-1 catalysts. A possible explanation consists in a beneficial effect of optimum availability of Fe^2+^ cation at pH 6.5 for plant growth [63]. This suggests that iron-containing clay-rich soils may mitigate the effect of ecotoxicity during the first stages of oxidative decomposition of organic pollutants. This still remains to be elucidated through deeper biological investigations.

The plant biomass amount (BA), often referred to as “Fresh Weight” (FW), was assessed as a third ecotoxicity criterion after 7 days exposure of *Lemna minor* individuals to ozonized mixtures. As compared to blank which contains no ozonized mixture, the general tendency is a BA increase for the first 5 min of ozonation of all organic substrates in agreement with increasing protein content (Figure 9). This BA increase was more pronounced when *Lemna minor* was in contact with ozonized DCF aliquots, and it was followed by a marked depletion even down to total disappearance for longer ozonation times of BPA aliquots in the presence of both catalysts at intrinsic pH.

The fact that all organic substrate mixtures ozonized in the presence Fe(II)Mt still produce high BA decay as compared to the blank suggests a key-contribution of iron (II) cations. Slightly weaker BA increase was noticed for DAZ-HMt-1 and BPA-HMt-1 both at pH 6.5, i.e., in the presence of iron-free clay catalyst. Much weaker but progressively depleting BA increase was registered in the present of DCF-HMt-1 at pH 6.5. These results are of great interest because they clearly demonstrate that the toxicity of ozonized media on *Lemna minor* is strongly influenced by the pH, the occurrence of iron cations, and compositions of both the clay catalyst and organic substrate.

Nonetheless, only lower BA values were obtained at instant pH, which decreases in times for all ozonized clay organic substrate mixtures. This confirms the detrimental effect and contribution to ecotoxicity of decreasing pH during organic pollutant oxidation, which are mainly reflected by visible decreases in the (Chl_A_/Chl_B_) ratio (Figure 10), protein content (Figure 11) and biomass amount (Figure 12).

The loss in correlation of all the three toxicity criteria at pH 6 and beyond must be due to an additional effect of change in iron (II) chemistry. Here, the occurrence of a pH threshold between direct ozonation with molecular ozone and radical generation process that may produce different intermediates distribution with different ecotoxicity should also be considered [2,41,54,55,57].

These findings are of great importance because they suggest the occurrence of an optimum pH for a normal growth of *Lemna minor* and that any slight deviation in the acidity of the medium affects plant metabolisms [64,65,66]. Deeper insights in this regard are expected to provide valuable data for better understanding the detrimental effect of partially oxidized organic molecules on biodiversity in clay-containing media.

## 3. Materials and Methods

### 3.1. Clay Materials Preparation

An acid-activated sample (HMt-1) was already prepared through 1 h impregnation of a crude bentonite with a Si/Al mole ratio of 2.50 supplied by Sigma Aldrich (2) in concentrated sulfuric acid [67,68]. A Na^+^-Montmorillonite-rich material (NaMt) with an Si/Al mole ratio of 2.47 and a cation exchange capacity (CEC) of 1.0 ± 0.05 meq.g^−1^ was obtained from previous bentonite purification involving repeated sedimentations and full ion-exchange with in aqueous NaCl solution [67,69,70]. For this purpose, 200 g of raw bentonite and 70 g of NaCl were slowly dispersed in 2000 mL of distilled water. The resulting suspension was then stirred at 80 °C for 7 h for achieving full clay swelling, then cooled down to room temperature and centrifuged 15 min in a Thermo Scientific ST16R Refrigerate Centrifuge (Waltham, MA, USA). The supernatant was removed and replaced with distilled water of the same volume followed by a clay suspension stirring for 4 h. This step was repeated until complete chloride disappearance based on AgNO_3_ test. The clay paste obtained by centrifugation was dried for 24 h at 30–40 °C and stored in a dry air enclosure [59,60,61]. CaMt, KMt and Fe(II)Mt samples were further prepared through consecutive ion exchange of 3 g of dry NaMt powder and 1 g of each corresponding salt (CaCl_2_, KCl, FeCl_2_) in 100 mL of distilled water. The mixture was stirred at 80 °C for 6 h, cooled and settled at room temperature for 24 h, then repeatedly washed with distilled water until the total disappearance of chloride. The resulting clay paste was air-dried for 24 h at 30–40 °C then stored in a dry air enclosure [68].

All clay materials investigated herein as adsorbents and catalysts were characterized through X-ray diffraction (Siemens D5000, CuKα, λ = 1.54051 Å) for crystallinity analysis. The determination of the Silicon/Aluminum ratio (Si/Al) was achieved through X-ray photoelectron spectrometry (XPS) measurements (PHI 5600-ci instrument, Physical Electronics, Eden Prairie, MN, USA) using an Al standard (1486.6 eV) as the anode for overflight spectra at 300 W. NaMt and ion-exchanged counterparts displayed no crystallinity loss, sharper 001 XRD reflexion and lower contents in volcanic ash and dense silica phases [45,62]. This accounts for a perfectly parallel arrangement of the clay lamellae resulting from bentonite purification and full ion-exchange. HMt-1 showed similar crystallinity as the starting bentonite and no structure collapse in spite of a decrease in Al content reflected by an increase of the Si/Al mole ratio from 2.50 to 2.69 [67].

### 3.2. Clay Behavior Study in Aqueous Media

The pH induced by each clay mineral in water and aqueous solutions of organic substrates was assessed using an Accumet^®^ model 15 device. The Zeta potential was determined by means of a ZetaPlus system (BrookHaven Instrument Corp., Holtsville, NY, USA, ZetaPlus/Bl-PALS). The average particle size was measured under optimized conditions (Viscosity 1.0031 cP, refractive index 1.33, absorption 0.1) in a 12 mm cell containing 2 mL of aqueous suspension of 2 mg L^−1^ of each catalyst previously prepared in nano pure water. This was achieved through the dynamic light scattering method (DLS) with a ZetaPlus particle sizer (Brookhaven Instruments) using 90 Plus Particles Sizing Software Version 4.20.

### 3.3. Adsorption and Ozonation Tests

A comprehensive study of the effect of the molecular structure was achieved with three probe molecules, namely bis-Phenol A (BPA), Diazinon (DAZ), and Diclofenac sodium (DCF) (Table 1). Atrazine (ATR) was only used for comparison in the adsorption studies. All the four organic compounds were purchased from Sigma-Aldrich. To prevent the effect of pH fluctuations induced by clay dispersion in the liquid media, adsorption was investigated at imposed pH values of 3; 6; 9; and 12 obtained by adding a few drops of 0.1 M H_2_SO_4_ or 0.1 M NaOH to 30 mL of each 30 mg L^−1^ solution of organic compound, further contacted with 40 mg of clay adsorbent at intrinsic pH and 21 ± 2 °C, under vigorous stirring for 10 min. A kinetic study was achieved by measuring the amount of adsorbate at equilibrium (***q_e_***) and at a time t (***q_t_***) for each probe molecule and applying the pseudo-first order and pseudo-second order models, respectively based on the absence or absence of chemical reactions.

In a second step, ozonation was conducted at room temperature in a 28 × 115 mm cylindrical glass reactor containing 30 mL of a Swedish Institute Standard (SIS) solution containing 30 mg L^−1^ of each probe molecule in the absence or presence of 40 mg of clay catalyst. The latter was previously ozonized for 5 min in pure distilled water for removing any trace of organic compounds that could contribute to ozone consumption during the oxidative degradation of the organic substrate. The reaction was run separately in different reactors under various exposure times to 600 mg h^−1^ bubbling from an A2Z ozone generator (Ozone Inc., Louisville, KY, USA).

### 3.4. Adsorption and Reaction Mixture Analysis

After centrifugation, the supernatants of both adsorption solutions and ozonation mixtures were qualitatively analyzed by UV-Vis spectrophotometry in the wavelengths range 190–800 nm in a 1 cm quartz cell. The device used was an Agilent-Cary 60 brand (Agilent Technologies, Santa Clara, CA, USA) equipped with a data processor. Quantitative assessment of both the adsorption and conversion yields was achieved by determining the residual amount of organic compound in the supernatant through liquid phase chromatography coupled with UV detection (HPLC-UV). For this purpose, an Agilent technology model 1200 brand instrument was used under previously optimized conditions (Appendix A) with a variable wavelength UV detector and a Star analysis software version 6.

### 3.5. Toxicitytests on Lemna minor

*Lemna minor* was supplied by Dr. Philippe Juneau laboratory (Biological Sciences Department, UQAM), and was cultivated according to a procedure of the Organization for Economic Cooperation and Development [46]. This culture was conducted in Pyrex flasks containing the growth medium of the Swedish Institute Standard (SIS) at a pH of 6.5 ± 0.2. The toxicity of the ozonized mixtures towards *Lemna minor* was evaluated by adding 4 three-fronds specimens of *Lemna minor* in the entire amount of mixture after adsorption or ozonation. For investigating the contribution of the acidity produced by ozonation to the toxicity, the tests were achieved at two different pHs: (1) at constant pH 6.5 considered as suitable for *L. minor* growth; here, the pH of the ozonized reaction mixture was adjusted by adding small amounts of aqueous 0.01 NaOH and (2) at the pH of the SIS mixture with the organic substrate solution before and after ozonation at different times. The observations were performed during 7 days with periodic assay tests for chlorophylls *a* and *b*, total proteins and biomass growth [47,48].

Chlorophyll was extracted according to Mazliak’s method. Thus, 50 mg of fresh material (leaves and roots) were ground in a mortar containing sand and then mixed with 5 mL of 80% acetone and a pinch of calcium carbonate to neutralize the vacuole acid (mixture of malic acid, oxalic acid, citric acid, and ascorbic acid). The resulting mixture was centrifuged for 10 min at room temperature. The determination of chlorophylls *a* (ChlA) and *b* (ChlB) and their total amount (ChlT) was achieved by UV-Vis spectrophotometry by measuring the absorbance (A) at 663 nm and 645 nm with a blank sample (80% acetone). The chlorophyll concentration is determined using the equations of Lichtenthaler and Buschman [59,60,61,62].

The total protein content of *Lemna minor* was assessed according to Bradford’s method after exposure to the ozonized mixture for 7 days. For this purpose, 50 mg of fresh material were taken from the cultivation medium, drained, and then ground in a mortar containing sand mixed with 5 mL of sodium phosphate buffer (0.1 mM at pH 7.0). After 15 min centrifugation, 30 µL of the ground material was mixed with 200 µL of Coomassie blue reagent and 770 µL of water, and then analyzed by UV-Vis spectrophotometry at 595 nm. A calibration curve was plotted in the range 0–60 µg mL^−1^ of bovine albumin previously prepared in potassium phosphate buffer (0.1 mM at pH 7.0). The amount of biomass was quantified through gravimetry using an analytical balance, after 7 days exposure of *Lemna minor* colonies to ozonized mixtures. The initial biomass was transferred into clean containers, rinsed with water, and then dried using absorbent paper [68].

## 4. Conclusions

This work allowed concluding that clay-catalyzed ozonation is a useful route to simulate the natural behavior of clay-containing media and to understand their self-regeneration. Adsorption contributes to the ozonation process, but layers of retained pollutant paradoxically promote hydrophobic Clay:Clay interaction and aggregation at the expense of the catalytic surface. This phenomenon is attenuated by surface cleaning upon pronounced ozonation, which however produces more acidity that reduces the surface charge density and interlamellar repulsion forces, thereby enhancing clay coagulation-flocculation. Effective ozonation requires a high extent of the adsorptive and catalytic surfaces through high clay dispersion. This can only be achieved by optimum hydrophilic-hydrophobic character and pH of the reaction mixture. The evaluation of three ecotoxicity criteria revealed the negative impact of incomplete oxidation of the organic pollutants on *Lemna minor*. Ecotoxicity arises mainly from both the acidic character and intrinsic toxicity of the yielded intermediates. These results provide essential foundations for correlating the degradability of organic pollutants to their molecular structure and the type of clay material and for predicting the ecotoxicity as a function of the host soils and neighboring waters.

## Data Availability

Not applicable.

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
