# Peer review of "Clay-Catalyzed Ozonation of Organic Pollutants in Water and Toxicity on Lemna minor: Effects of Molecular Structure and Interactions"

_molecules, 2022, doi:10.3390/molecules28010222_

Round 1

Reviewer 1 Report

The suggested paper by Wembe et al. entitled: Clay-catalyzed ozonation of organic pollutants in water and toxicity on Lemna minor- Effects of molecular structure and interactions. First of all, the idea is good, simple and attractive for researchers. The overall topic of this manuscript is of potential significance for adsorption and ozonation of organic pollutants, which will add effective impact for environmental sustainability. The article discussed the scientific details clearly. To my opinion, I will recommend the acceptance of this manuscript with a minor revision. Please check the following comments:

1.      Please check typo mistakes throughout the manuscript.

2.      The use of catalyst in adsorption process is not correct scientifically, the word adsorbent is more accurate.

3.      What about recyclability of the materials? It will be good if reusability of materials tested toward adsorption and ozonation processes.

Author Response

Answer to reviewer 1

Reviewer #1:

Reviewer’s general comments

The suggested paper by Wembe et al. entitled: Clay-catalyzed ozonation of organic pollutants in water and toxicity on Lemna minor- Effects of molecular structure and interactions. First of all, the idea is good, simple and attractive for researchers. The overall topic of this manuscript is of potential significance for adsorption and ozonation of organic pollutants, which will add effective impact for environmental sustainability. The article discussed the scientific details clearly. To my opinion, I will recommend the acceptance of this manuscript with a minor revision. Please check the following comments:

Reviewer’s specific comments

  1. Please check typo mistakes throughout the manuscript.

Authors’ response

Done. A full spelling of the text has been performed and some style issues have been addressed. Changes are marked in green color throughout the whole revised manuscript.

Reviewer’s specific comments

  1. The use of catalyst in adsorption process is not correct scientifically, the word adsorbent is more accurate.

Authors’ response

Done. Yes, indeed. For adsorption, this is an abusive language issue that has been addressed and some similar mistakes have been corrected accordingly. Changes are marked in green color throughout the whole revised manuscript.

Reviewer’s specific comments

  1. What about recyclability of the materials? It will be good if reusability of materials tested toward adsorption and ozonation processes.

Authors’ response

Done. These clay minerals are natural materials that already occur in most soils. They also contain harmless trace amounts of metal cations (Na+ and Fe2+) that also occur in most soils. Bentonites and their deriving montmorillonites are low cost and widely available. Their reuse is as convenient and possible as their very preparation, but their reusability is needed only if justified. Their use is this research was intended to simulate the natural behavior of clay-containing media and their self-regeneration (Soils and turbid aquatic media). Similar statements were already provided in the introduction sections.

However to comply with this reviewer’s comment, some the above sentence were inserted before the last paragraph of the introduction section.  Changes are marked in green color where made.

Reviewer’s specific comments

The reviewer suggested in his table an improvement of the conclusion section as an answer to the question: Are the conclusions supported by the results?

Authors’ response

Done. The conclusion has been thoroughly reformulated. Changes are marked in green color where made.

Reviewer 2 Report

In this paper, the authors have reported the catalytic ozonation of organic pollutants using clay as a catalyst. The results are interesting. However, the idea of the manuscript is not novel. Clay is a well known catalyst and there are several reports of its applications in various catalytic processes. On the other hand, ozonation is a known reaction for the removal of pollutants. All in all, I did not find the manuscript novel for publication in Molecules. Therefore, I recommend the rejection of the manuscript. 

Author Response

Answer to reviewer 2

Reviewer #2:

Reviewer’s general comments

In this paper, the authors have reported the catalytic ozonation of organic pollutants using clay as a catalyst. The results are interesting. However, the idea of the manuscript is not novel. Clay is a well-known catalyst and there are several reports of its applications in various catalytic processes. On the other hand, ozonation is a known reaction for the removal of pollutants. All in all, I did not find the manuscript novel for publication in Molecules. Therefore, I recommend the rejection of the manuscript. 

Authors’ response

Indeed, clay-catalyzed ozonation is not a novelty, and it has even become an almost well-known process. Nevertheless, this was not the main scope of the present research. Here, this process was only used to simulate the natural behavior of clay-containing media (Soils and turbid aquatic media) for a better understanding of their self-regeneration.

The main target of the present work resided in demonstrating that the natural oxidative degradation of organic pollutants and ecotoxicity of oxidized mixtures can be correlated to their molecular structure and type of clay adsorbent-catalysts. This is expected to provide a tool for predicting the ecotoxicity level of organic pollutants released in nature (Here, only for a given living species, of course) according to the host clay-containing media.

For the sake of clarity, this last formulation of the main objective will replace the previous one in the penultimate paragraph of the introductory section. See changes marked in green.

Besides, given the lack of precise comments, we granted a special attention to the reviewer suggestions for the manuscript improvement in his table by responding to the raised questions, namely

Question in the first table of the reviewer’s report

Moderate English changes required

Authors’ response

Done. A full spelling of the text has been performed and some style issues have been addressed to improve the English standard. Changes are marked in green color where made throughout the whole revised manuscript.

Question in the 2nd table of the reviewer’s report

Does the introduction provide sufficient background and include all relevant references?

Reviewer’s answer: Can be improved.

Authors’ response

Done. Some sentences have been rephrased and others were added in order to include new references in the introduction section.

Question in the 2nd table of the reviewer’s report

Are the methods adequately described?

Reviewer’s answer: Can be improved.

Authors’ response

Done. Some sentences has been rephrased and/or enriched with additional details in order to improve the method description. See changes marked in green color in the experimental section.

Question in the 2nd table of the reviewer’s report

Are the results clearly presented?

Reviewer’s answer: Can be improved.

Authors’ response

Done. For the sake of clarity, Figure 2 has been modified, a related discussion has been added before and after this figure and some confusing sentences have been reformulated. See changes marked in green color in the section devoted to the ‘’Results and discussion’’.

Reviewer 3 Report

The authors have organized the experimental section in a proper and solid structure.

Nevertheless, some minor revisions should be considered prior to publication: 
- A total percentage of self-citations are detected for the author Azzouz (36%=19/52). Either reduce the self-citations or add some relevant literature intext. The total self-citation percentage must be about 25%.

Perhaps some references in the overall pharmaceutical treatment procedures should be reported. A small paragraph should be appropriate.

Sentences 44-54 should be more enriched with literature. The authors could elaborate more.

In Figure 1 please change the signs within figures. eg circles for ATR Fe(II)Mt and squares for ATR HMt-1H. Please repeat as many Figures as necessary.

In Figure 2 please add a distinctive trendline to supoprt the tendency of the zeta potential

Author Response

Answer to reviewer 3

Reviewer #3:

Reviewer’s general comments

Comments and Suggestions for Authors: The authors have organized the experimental section in a proper and solid structure. Nevertheless, some minor revisions should be considered prior to publication: 

Authors’ response

Done. All the raised issues requiring minor revisions have been addressed. See their point-to-point responses are provided below.

Reviewer’s specific comments

- A total percentage of self-citations are detected for the author Azzouz (36%=19/52). Either reduce the self-citations or add some relevant literature in text. The total self-citation percentage must be about 25%.

Perhaps some references in the overall pharmaceutical treatment procedures should be reported. A small paragraph should be appropriate.

Authors’ response

Done. The number of self-citations has been reduced as much possible, because all references previously self-cited were strongly necessary to support the knowledge core on clay-catalyzed ozonation of organic molecules.

Reviewer’s specific comments

Sentences 44-54 should be more enriched with literature. The authors could elaborate more.

Authors’ response

Done. New references have been added and the corresponding sentences have been slightly rephrased.

Reviewer’s specific comments

In Figure 1 please change the signs within figures. eg circles for ATR Fe(II)Mt and squares for ATR HMt-1H. Please repeat as many Figures as necessary.

Authors’ response

Done. The signs for the experimental data have been changed in figures 1 to 3.

Reviewer’s specific comments

In Figure 2 please add a distinctive trend line to support the tendency of the zeta potential

Authors’ response

Done. A new trend line has been added to highlight the ZP tendency in figure 2a. Another trend line has been inserted in the new figure 2b and a related discussion has been added at the end of page 9. See the new paragraph colored in green before and after figure 2.

Question in the 2nd table of the reviewer’s report

The reviewer suggested in his table an improvement of the reference section as an answer to the question: Are all the cited references relevant to the research?

Reviewer’s answer: Can be improved.

Authors’ response

Done. New references have been added in the introduction section where some paragraphs have been added and other more or less slightly reformulated.

Round 2

Reviewer 2 Report

I see the authors have improved the quality of their paper in the revised version.